# Stratify or Inject: Two Simple Training Strategies to Improve Brain Tumor Segmentation

**Raphael Meier**[1]                                                    raphael.meier@insel.ch
**Michael Rebsamen**[1]                                      michael.rebsamen@insel.ch
**Urspeter Knecht**[2]                                      urspeter.knecht@istb.unibe.ch
**Mauricio Reyes**[2,3]                                mauricio.reyesaguirre@insel.ch
**Roland Wiest**[1]                                                  roland.wiest@insel.ch
**Richard McKinley**[1]                              richardIain.mckinley@insel.ch

[1] *Support Center for Advanced Neuroimaging (SCAN), University Institute of Diagnostic and Interventional Neuroradiology, University of Bern, Inselspital, Bern University Hospital, Bern, Switzerland*

[2] *Institute for Surgical Technology and Biomechanics, University of Bern, Bern, Switzerland*

[3] *Healthcare Imaging A.I. Lab, Insel Data Science Center, Inselspital, Bern University Hospital, Bern, Switzerland*

**Editors:** Under Review for MIDL 2019

## Abstract

Deep learning methods for brain tumor segmentation are typically trained in an ad hoc fashion on all available data. Brain tumors are tremendously heterogeneous in image appearance and labeled training data is limited. We argue that incorporation of additional prior information, specifically tumor grade, associated with tumor imaging phenotypes during model training can significantly improve segmentation performance. Two strategies for incorporation of tumor grade during model training are proposed and their impact on segmentation performance is demonstrated on the BRATS 2018 dataset.

## 1. Introduction

The segmentation of brain tumors has been a long standing problem in medical image analysis. Research on this topic has been accelerated greatly through the availability of public datasets such as the Brain Tumor Segmentation (BRATS) Challenge dataset (Menze et al., 2015; Bakas et al., 2017a,b,c). Currently, the best performing methods for brain tumor segmentation are based on deep learning (Bakas et al., 2018), with first approaches being applied in clinically critical areas such as tumor response assessment (Kickingereder et al., 2019) or radiation therapy planning (Jungo et al., 2018). A general view in deep learning is that more training data yields better generalization performance. For tasks in computer vision it was shown that model performance increases logarithmically based on volume of training data (Sun et al., 2017). Consequently, deep learning segmentation models are trained on all available data, often neglecting peculiarities of the data at hand.

The BRATS Challenge has been concerned so far with the segmentation of glioma, which are primary tumors of the central nervous system. Glioma can be classified into different tumor grades based on the underlying molecular characteristics and histology (Louis et al.,

2016). A higher grade reflects increasing malignancy and glioma are commonly grouped into high-grade (grade III/IV) and low-grade glioma (grade I/II). Furthermore, they exhibit a tremendous genetic and molecular heterogeneity which spans across tumor grades but also manifests itself within a particular type such as glioblastoma (grade IV) (Verhaak et al., 2010; Sottoriva et al., 2013). The underlying biological configuration of a tumor has been associated with distinctively different tumor imaging phenotypes (Grossmann et al., 2016). In general, low-grade glioma present much less or no contrast-enhancement compared to high-grade glioma (Forst et al., 2014). Deep learning methods are confronted with the challenge to successfully generalize across these different imaging phenotypes.

We hypothesize that brain tumor segmentation performance of deep learning methods can be improved by utilizing prior information associated with tumor imaging phenotypes during model training. Thus, we propose two simple training strategies targeted at tumor grade and evaluate their effectiveness on the BRATS 2018 dataset using a recently proposed, top-ranked method (McKinley et al., 2019). This work is part of a more extensive study currently being submitted to a journal.

## 2. Methods

**Model architecture.** The deep learning method corresponds to a shallow U-Net style model of down and upsampling connections featuring densely connected blocks of dilated convolutions. For more details on the model architecture we refer to (McKinley et al., 2019).
**Incorporation of tumor grade.** We propose two strategies to utilize information on tumor grade at the stage of model training. The first strategy consists of *stratifying* the training data into high-grade glioma (HGG) and low-grade glioma (LGG) cases and training two separate models. During testing the respective model is applied to testing data with corresponding tumor grade. As a second strategy, we propose an *injection* of the tumor grade. In addition to feeding the model with imaging data consisting of the co-registered Magnetic Resonance (MR) sequences, we provide it with a binary input indicating if the case at hand is either a LGG or HGG. The tumor type is injected as an image volume, with dimensions identical to the MR sequences and all voxel values set either to zeros or ones. The model is then trained on this enlarged dataset (type-aware network). For both strategies, the model architecture and hyperparameter setting remains unchanged.

In the following, we utilized training data of the BRATS 2018 Challenge, which includes 75 patients with LGGs and 210 patients with HGGs. We did not include the testing data in the analysis since tumor grade is blinded for those cases. A detailed description of the imaging data can be found in (Bakas et al., 2018). Four different models were trained using five-fold cross-validation: i) a baseline model using all available cases (N=285), ii) a model trained only on HGG data (N=210), iii) a model trained only on LGG data (N=75), and iv) the type-aware network trained on all cases (N=285).

## 3. Results & Conclusion

The segmentation performance of the four different deep learning models in terms of fraction of cases with improved Dice coefficient is shown in Table 1. If we look at the HGG data alone, the HGG model yields a significantly improved performance over the baseline model

Table 1: Ratio in % of better performing subjects compared to baseline. p-values from a one-sided Wilcoxon signed rank test. Bold numbers indicate statistically significant ($p < 0.05$) results. (CE = contrast-enhancing tumor)

|  | CE | Core | Tumor |
|---|---|---|---|
| LGG vs. Baseline | 41.7 (p=0.877) | 49.3 (p=0.454) | 54.7 (p=0.208) |
| HGG vs. Baseline | **58.4 (p=0.005)** | **70.3 (p=5.659e-09)** | 46.7 (p=0.877) |
| HGG/LGG vs. Baseline | 54.6 (p=0.127) | **64.9 (p=1.441e-05)** | 48.8 (p=0.725) |
| Type-aware vs. Baseline | 53.4 (p=0.321) | **53.9 (p=0.028)** | 52.6 (p=0.231) |

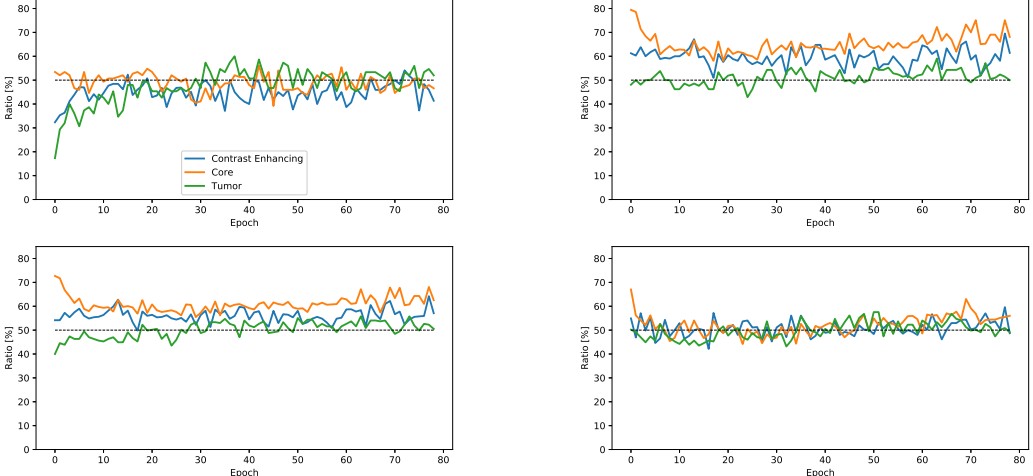

Figure 1: Impact of proposed strategies on model performance over all epochs. From top left to bottom right: LGG vs. Baseline, HGG vs. Baseline, HGG/LGG vs. Baseline, Type-aware network vs. Baseline.

in 58.4% of the cases for contrast-enhancing tumor and 70.3% of the cases for the tumor core segmentation. Looking at all the data including also LGG, we see a significant improvement for the tumor core segmentation in 64.9% of the cases when using the two models trained on stratified data compared to the baseline model trained on all available data. Finally, the type-aware network yields a minor but significant improvement for the segmentation of the tumor core in 53.9 % of the cases. Figure 1 shows that observed improvements are manifested relatively consistent across all epochs of model training.

We have proposed two different strategies on incorporating information of tumor grade during model training, which are straightforward to apply, and demonstrated their effectiveness on the BRATS 2018 dataset. While data stratification yields clear improvements, more advanced network architectures incorporating prior information about tumor grade beyond injecting it as an additional input should be investigated further. In addition, our strategies could also be used in conjunction with networks for tumor typing.

## Acknowledgments

This work was supported by the Swiss National Foundation, grant number 169607.

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
