# OpenReview forum: "Stratify or Inject: Two Simple Training Strategies to Improve Brain Tumor Segmentation"
_MIDL.io/2019/Conference/Abstract — MIDL Abstract 2019_

### Official Review · AnonReviewer2 · 2019-04-25
**interesting empirical evaluation**

**Rating:** 3
**Confidence:** 3

**Review:**

This straight-to-the-point paper compares two approaches to incorporating known phenotype into a tumor segmentation CNN: stratifying (i.e., training two separate models) vs. injecting the (binary) class in the net. If I may make a suggestion: in the extended version, it would be very interesting to compare performance as a function of training cases (one would thing that for high N stratification may be better) and also evaluate other injection strategies.

---

### Official Review · AnonReviewer1 · 2019-04-30
**Interesting strategy to improve tumor segmentation**

**Rating:** 3
**Confidence:** 2

**Review:**

The abstract presents two strategies to improve segmentation on BRATS dataset. The paper trains four individual models with five-fold validation. The results are reported as a ratio of better performing subjects to baseline.
It would be interesting to know in the extended version of the paper the following,
*Are the results improving for the model trained only on HGG as the split contains more data compared to LGG?
*What are the actual segmentation results in terms of accuracy or Dice. Knowing this would benefit in assessing the importance of the strategy?
*The ratio of better performing subjects does not directly correlate to improvement in segmentation performance. It is also possible that when the model performs better for 60% of subjects, it could fail badly for the 40% resulting in a decreased segmentation performance.

---

### Decision · Program_Chairs · 2019-05-06
**Acceptance Decision**

Accept